# Modulating Cholesterol Metabolism via ACAT1 Knockdown Enhances Anti-B-Cell Lymphoma Activities of CD19-Specific Chimeric Antigen Receptor T Cells by Improving the Cell Activation and Proliferation

**DOI:** 10.3390/cells13060555

**Published:** 2024-03-21

**Authors:** Qiong Su, Jie Yao, Muhammad Asad Farooq, Iqra Ajmal, Yixin Duan, Cong He, Xuefei Hu, Wenzheng Jiang

**Affiliations:** Shanghai Key Laboratory of Regulatory Biology, School of Life Sciences, East China Normal University, Shanghai 200241, China

**Keywords:** chimeric antigen receptor, CAR-T, ACAT1, B-cell lymphoma, immunotherapy

## Abstract

CD19-specific CAR-T immunotherapy has been extensively studied for the treatment of B-cell lymphoma. Recently, cholesterol metabolism has emerged as a modulator of T lymphocyte function and can be exploited in immunotherapy to increase the efficacy of CAR-based systems. Acetyl-CoA acetyltransferase 1 (ACAT1) is the major cholesterol esterification enzyme. ACAT1 inhibitors previously shown to modulate cardiovascular diseases are now being implicated in immunotherapy. In the present study, we achieved knockdown of ACAT1 in T cells via RNA interference technology by inserting ACAT1-shRNA into anti-CD19-CAR-T cells. Knockdown of ACAT1 led to an increased cytotoxic capacity of the anti-CD19-CAR-T cells. In addition, more CD69, IFN-γ, and GzmB were expressed in the anti-CD19-CAR-T cells. Cell proliferation was also enhanced in both antigen-independent and antigen-dependent manners. Degranulation was also improved as evidenced by an increased level of CD107a. Moreover, the knockdown of ACAT1 led to better anti-tumor efficacy of anti-CD19 CAR-T cells in the B-cell lymphoma mice model. Our study demonstrates novel CAR-T cells containing ACAT1 shRNA with improved efficacy compared to conventional anti-CD19-CAR-T cells in vitro and in vivo.

## 1. Introduction

Efforts over the last forty years to develop novel approaches to treat B-cell lymphoma have proven successful. While most individuals diagnosed with diffuse large B-cell lymphoma (DLBCL) can achieve a cure through the conventional rituximab, cyclophosphamide, doxorubicin, vincristine, and prednisone (R-CHOP) treatment, those who do not respond to R-CHOP face a bleak prognosis [1,2,3]. In cases of relapse after first-line therapy, platinum-based salvage chemotherapy has been used as a second-line therapy but has proved to have a poor prognosis with a median survival of about 6 months [4,5,6,7,8]. Recently, four second-generation chimeric antigen receptor (CAR) T cell therapy products tisagenlecleucel, axicabtagene ciloleucel, brexucabtagene autoleucel, and lisocabtagene maraleucel targeting CD19 antigen have been approved by the FDA for the treatment of B-cell lymphoma [9,10].

CD19 is a transmembrane protein that takes part in regulating the activation of B-cells in an antigen-dependent fashion. CD19 is expressed uniformly in every stage of B-cell differentiation and is also passed during malignant transformation. In more than 95% of B-cell malignancies, CD19 is expressed [11], whereas, hematopoietic stem cells do not express CD19, hence making it an excellent target in B-cell pathologies without affecting other hematopoietic lineages [12].

CARs expressing co-stimulatory domain 4-1BB have shown great efficacy in the acute lymphoblastic leukemia xenograft model, and they can produce activity that is antigen-independent, which contributes to their enhanced activity in vivo. In vitro studies also demonstrate that the integration of 4-1BB signaling domains boosts CAR response more proficiently than the CD28 domain [13].

Acetyl-CoA acetyltransferase (ACAT) in the human body catalyzes the conversion of free cholesterol to cholesterol esters or lipid droplets, which are then deposited in the cell cytoplasm [14]. In mammals, there are two subtypes of ACAT, ACAT1 and ACAT2. A recent study indicates the expression of ACAT1 in activated T cells, while ACAT2 was expressed in negligible amounts. Activated CD8^+^ T cells synthesize more free cholesterol to support rapid cell proliferation. Moreover, there is evidence of the involvement of cholesterol in CD8^+^ T cell signaling. It has been reported that the plasma membrane cholesterol level of T cells was upregulated by inhibiting ACAT1 and hence, favored T-cell signal transduction [15,16]. ACAT1 ablation in CD8^+^ T cells leads to increased production of cytotoxic and pro-inflammatory cytokines like IFN-γ, TNF-α, and GzmB [17].

In the present study, we have screened an effective shRNA to knockdown ACAT1 and constructed a lentiviral vector that expressed the shRNA-ACAT1 gene and the second-generation CAR molecule targeting CD19. Anti-CD19-CAR-T cells with silencing of ACAT1 exhibited an enhanced ability to kill B-cell lymphoma in vitro and in vivo.

## 2. Materials and Methods

### 2.1. Cell Lines

HEK293T, Jurkat, Raji, and Daudi cell lines were obtained from ATCC. All cells were maintained in either DMEM (10% FBS and 1% penicillin/streptomycin) or RPMI 1640 (10% FBS and 1% penicillin/streptomycin) media.

### 2.2. Antibodies and Reagents

T4 DNA Ligase and QuickCut^TM^ HpaⅠ, XhoⅠ, EcoRⅠ, and AfeⅠ were obtained from NEB (Ipswich, MA, USA). RNA Extraction Kit, Reverse Transcription Kit, SYBR Premix EX Taq^TM^, and PrimeSTAR Max DNA Polymerase were purchased from Takara Biomedical Technology (Beijing) Co., Ltd. (Beijing, China) SDS-PAGE Gel Rapid Preparation Kit was obtained from Shanghai Yeasen Biotech Co., Ltd. (Shanghai, China) and Easy II protein quantification kit was purchased from Beijing Quanshijin Biologivasl Co., Ltd. (Beijing, China) High Glucose DMEM Medium and RPMI 1640 Cell Culture Media were GIBCO products (Carlsbad, CA, USA). X-VIVO^TM^ 15 medium was obtained from Lonza (Basel, Switzerland). Ficoll-Paque for T-cell isolation was obtained from HyClone, Logan, UT, USA. Human IL-7, IL-15, and IL-21 were purchased from Peprotech (Rocky Hill, CT, USA). CD4 and CD8 microbeads were purchased from Miltenyi Biotech (Bergisch Gladbach, Germany). Anti-human CD19-Cy5.5 (Cat # 561295) flow cytometry antibody was obtained from BD Biosciences (Franklin Lakes, NJ, USA). Anti-human CD69-APC (Cat # 310910), Anti-human IFN-γ-APC (Cat # 502512) Anti-human GzmB-PE/Cy7 (Cat # 372214), Anti-human CD107a-PE/Cy7 (Cat # 328618) and Annexin V-APC (Cat # 640920) were purchased from BioLegend (San Diego, CA, USA).

### 2.3. Construction of Lentiviral Expression Plasmid and Lentivirus Production

mRNA sequence of the ACAT1 gene (Accession: NM_000019) was attained from NCBI Genbank database. Three ACAT1-shRNA sequences and a negative control (NC) sequence were randomly selected from the website of BLOCK-iT^TM^ RNAi designer. These selected sequences lie in the Open reading frame (ORF) of the gene, have a high GC content, and their 5′ end started from the “G” nucleotide. The sequences along with their GC content are as follows: shACAT1-A (5′ GCTTGGTTCCATTGCAATTCA 3′, 42.6%), shACAT1-B (5′ GGAAATAAGATATGTGGAACG 3′, 38.1%), and shACAT1-C (5′ GGTGCAGGAAATAAGATATGT 3′, 38.1%).

shRNA were ligated into a pLL3.7 vector containing U6 promoter and EGFP. Chimeric antigen receptor containing scFv of CD19, 4-1BB co-stimulatory domains, and CD3ζ chain was cloned into the downstream of EF1αpromoter in pLL3.7 lentivirus backbone to generate CD19-CAR plasmid. After selecting the efficient shRNAs, another lentiviral pLL3.7 vector encoding CD19-CAR in combination with the ACAT1-shRNA sequence or shRNA-NC sequence was constructed. All of the plasmids were verified by restriction digestion method and further confirmed by sequencing analysis. Lentiviruses were produced by transient transfection of HEK293T cells as described previously [18]. Harvested lentiviral supernatant underwent ultracentrifugation and was stored at −80 °C.

### 2.4. T Cell Sorting and Activation

Fresh peripheral blood mononuclear cells (PBMCs) were collected under a protocol approved by the Ethics Committee of East China Normal University, following written informed consent. PBMCs were isolated from 200 mL blood by using the Ficoll-Paque PLUS density centrifugation method. Different layers were obtained after centrifugation at 1300 rpm for 10 min. The upper plasma layer was removed and the second lymphocyte layer was carefully collected and transferred into another 50 mL conical tube. A total of 15 mL of lymphocyte separation solution was added and then centrifuged at 800 rpm. The supernatant containing platelets was carefully aspirated and the pellet was resuspended in PBS. After washing twice with PBS, the cells were counted. PBMCs (10^7^) were stained with 20 μL of CD3 magnetic beads and incubated for 15 min in the incubator. After T cells were isolated, CD4 and CD8 microbeads were used for T-cell activation. Following 48 h of activation, T cells were cultured in X-VIVO^TM^ 15 medium containing IL-7, IL-15, and IL-21.

### 2.5. Cytotoxic Assay

To assess the anti-tumor effect of shACAT1-19CAR-transduced T cells in comparison to CD19-CAR-transduced T cells, CAR-T cells were co-cultured with 2 × 10^4^ Raji or Daudi cell lines. Target and effector cells were seeded in a 96-well plate at an E:T ratio of 1:1. After 16 h of culture, cells were harvested and stained with anti-human CD19-PE antibody. CD19-positive cells were measured using a flow cytometer and the percentage of cytotoxicity was calculated.

### 2.6. Cell Activation, Degranulation, and Cytokine Release Analysis

Non-transduced and CD19-CAR- and shACAT1-19CAR-transduced T cells were co-cultured with Raji cell lines at an E:T ratio of 1:1 for 4 h in a 24-well plate. After incubation, the co-cultured cells were washed twice with staining buffer and stained with anti-human CD69-APC. The samples were analyzed by flow cytometry. After 16 h of co-culture, cells from experimental cohorts were collected, washed, and either resuspended in 10 μL fixation and permeabilization solution at 4 °C overnight or stained with anti-human CD107a-PE/Cy7 antibody to measure the CD107a expression. Fixed cells were washed with 1× washing solution and stained with anti-human IFN-γ-APC and anti-human GzmB-PE/Cy7 in separate experiments for IFN-γ and GzmB analysis, and subjected to flow cytometry.

### 2.7. Cell Proliferation Assay

To evaluate the effect of ACAT1 silencing on CD19-CAR-transduced T-cell proliferation, 5(6)-carboxyfluorescein diacetate succinimidyl ester (CFSE) staining was used. T cells (1 × 10^6^) were infected with lentiviruses expressing CD19-CAR and shACAT1-19CAR and were stained with 2 μM/mL CFSE staining for 10 min in a 37 °C incubator. To stop the reaction, 5 mL RPMI medium (with 10% FBS and 1% P/S) and complete X-VIVO medium were added to the cells, which were then placed at 4 °C for 10 min. After the cells were cultured at indicated days, the cells were collected and rinsed twice with PBS. CSFE dilution was analyzed by flow cytometer on day 0 and day 5.

### 2.8. qPCR

To evaluate the interference effect of ACAT1 in Jurkat cells or primary T cells, qPCR was performed. Jurkat or primary T cells (1 × 10^6^) were seeded in a 24-well plate and infected by lentiviruses expressing ACAT1-shRNAs. After 2 days of infection, the cells were collected and washed with PBS twice. A total of 1 mL RNAiso Plus was added and total RNAs were extracted according to the protocol. mRNAs were then reverse transcribed into cDNA according to the Reverse Transcription Kit instructions and qPCR was performed with ACAT1-specific primers with GAPDH used as a reference.

### 2.9. Western Blot

Jurkat cells or primary T cells (1 × 10^6^) were infected with lentiviruses expressing ACAT1-shRNAs and the cells were collected after 2 days of infection. Cells were washed with ice-cold PBS then lysed in 200 μL cold RIPA lysis buffer and incubated in ice for 30 min. After centrifugation at 12,000 rpm for 15 min at 4 °C, the protein concentration was determined by Easy II Protein Quantification Kit. Samples were denatured at 100 °C for 10 min and electrophoresis was performed on 10% SDS-PAGE. After being transferred to the nitrocellulose membrane, ACAT1 protein was detected using an antibody (Cat # 44276, Cell signaling technology, MA, USA or Cat # ab154396, Abcam, Cambridge, UK) and visualized. β-actin (Cat# 4967, Cell signaling technology) or GAPDH (Cat# 14C10, Cell signaling technology) was used as a reference.

### 2.10. Xenogeneic Lymphoma Models

To evaluate the in vivo anti-tumor effect of transgenic CD19-CAR and shACAT1-19CAR T cells, female 6- to 8-week-old NOD/SCID/γ-chain^−/−^ (NSG) mice were used. Mouse experiments were performed following the guidelines of the Animal Ethics Committee of East China Normal University. Briefly, mice were injected subcutaneously with 3 × 10^5^ Raji cancer cells with Luciferase gene, and IVIS imaging was performed on the 6th day, post-tumor. CD19-CAR (5 × 10^6^) and shACAT1-19CAR T (5 × 10^6^) cells were injected in mice on the 7th day and weekly IVIS imaging was performed until the 34th day, post-tumor inoculation.

### 2.11. Statistical Analysis

One-way or two-way ANOVA was used to determine the statistical significance of differences between samples, and *p* < 0.05 was accepted as indicating a significant difference. Survival was plotted using a Kaplan−Meier survival curve and statistical significance was determined by the Log-rank (Mantel−Cox) test. Prism software version 8.0 (GraphPad, La Jolla, CA, USA) was used for statistical analysis.

## 3. Results

### 3.1. Screening of shRNA to Silence ACAT1 Expression

Downregulation of ACAT1 results in reduced cholesterol esters formation and produces greater levels of free cholesterol in the cell membranes of T lymphocytes. RNA interference has been used as a tool to regulate and silence the expression of specific genes [19]. To knockdown the ACAT1 gene, three different ACAT1 shRNAs (shACAT1-A, shACAT1-B, and shACAT1-C) and negative control (shRNA-NC) were designed, and shRNA oligonucleotide was inserted into the pLL3.7 lentiviral vector backbone as explained in the schematic diagram in Figure 1a. Successful insertion of ACAT1 shRNAs into pLL3.7 lentiviral vector was confirmed by restriction digestion and further by sequencing.

HEK293T cell line was coinfected with either shACAT1-A, shACAT1-B, or shACAT1-C plasmids in the presence of packaging plasmid (psPAX-2 and pMD2.G) to generate respective lentiviruses termed LV-shACAT1-A, LV-shACAT1-B, LV-shACAT1-C, and LV-shRNA-NC. More than 8 × 10^6^ TU/mL virus titer was observed in all four shRNA-expressing lentiviruses and was used directly to transduce Jurkat cells yielding 99% efficiency (Appendix A). After 48 h, the cells were collected and subjected to RNA extraction and subsequent cDNA synthesis. The qPCR data demonstrated that all three shRNAs successfully downregulated ACAT1 gene expression in transduced Jurkat cells. Western blot analysis was carried out to confirm the qPCR data and a similar trend was seen (Figure 1b,c). The gene and protein expression data revealed that shACAT1-A and shACAT1-C demonstrated better knockdown efficiency than shACAT1-B. Furthermore, based on the chosen shRNA sequences, we selected shACAT1-A for further experiments due to its high GC content. The higher GC content signifies a stable secondary structure and effective interaction with target mRNA. To check the persistent downregulation efficiency, we performed qPCR analysis at different time intervals in shACAT1-A-transduced Jurkat cells. The shACAT1-A provided stable downregulation until 8 days and was selected for further analysis (Figure 1d).

### 3.2. Silencing of ACAT1 Enhances Anti-B-Cell Lymphoma Activity of CD19-CAR-Transduced T Cells

After confirming the knockdown efficiency in Jurkat cells, shACAT1-A was inserted into CD19-targeting CAR (19CAR), and the resultant plasmid was termed shACAT1-19CAR (Figure 2a). Primary T cells were then transduced with either 19CAR, NC-19CAR, or shACAT1-19CAR lentiviruses to generate respective CAR-T cells. The cells from experimental cohorts were then collected after 2 days and subjected to Western blot analysis for ACAT1 expression. The results indicated that ACAT1 expression was reduced significantly in shACAT1-19CAR-T cells (Figure 2b). The expression of CD19 on Raji and Daudi cell lines was detected by flow cytometry analysis. The data showed that there was high-level expression of CD19 on Raji and Daudi cell lines and they could be used as the target cells in our study (Appendix A). We further analyzed the cytotoxic potential of our novel CAR-T cells against B-cell lymphoma cell lines. The non-transduced T cells, 19CAR-T, NC-19CAR-T, and shACAT1-19CAR-T cells were co-cultured with target cells and subsequently analyzed by FACS. As both the Raji and Daudi cells highly expressed CD19, the expression of CD19 was determined by flow cytometry as a measure of living target cells. The results indicated that CD19-CAR-T cells could efficiently kill Raji cells and Daudi cells, while silencing of ACAT1 gene expression could significantly enhance the cytotoxic capacity of CD19-CAR-T cells (Figure 2c,d).

### 3.3. Silencing of ACAT1 Gene Enhances the Cell Activation and Degranulation of CD19-CAR-Transduced T Cells

Several proteins are known to be upregulated during the immune activation process. CD69 is one of the most commonly studied activation markers due to its early expression in activated T, B, and NK cells. After T-cell activation, the expression levels of a large number of effector molecules such as IFN-γ and granzyme B (GzmB) are upregulated as a result of degranulation. In the current study, we have used the CAR-T cells that contain both CD4^+^ and CD8^+^ cells and were co-cultured with Raji cells to analyze the expression of the surface or intracellular molecules by flow cytometry. In our study, ACAT1 knockdown significantly upregulated CD69 expression, indicating that the activation of 19CAR-T cells was improved (Figure 3a). Additionally, the experimental cohorts were stained with antihuman IFN-γ antibodies and analyzed by FACS. shACAT1-19CAR-T cells exhibited increased proportions of IFN-γ expressing cells compared to the wild-type CAR-T cells (Figure 3b). The enhanced activation and cytokine release in shACAT1-19CAR-T cells might be due to the modulation of membrane properties and increased responsiveness of T cells to tumor antigens as a result of ACAT1 knockdown. The ability of CAR-T cells to express GzmB and degranulation marker CD107a was also improved upon ACAT1 downregulation (Figure 3c,d). Overall, the data indicated that shACAT1-19CAR-T cells outperformed 19CAR-T cells in terms of activation, degranulation, and cytokine release, all of which are important parameters of T cell cytotoxic potential.

### 3.4. Silencing of ACAT1 Gene Enhances the Cell Proliferation of CD19-CAR-T Cells

It is reported that the inhibiting activity of the key cholesterol esterification enzyme ACAT1 can upregulate the plasma membrane cholesterol level of T cells, which leads to enhanced TCR clustering and signaling as well as more efficient formation of the immunological synapse [15]. To address whether downregulation of ACAT1 expression affected the cell proliferation of CD19-CAR-T cells in vitro, CD19-CAR-T cells were stained with 5(6)-carboxyfluorescein diacetate succinimidyl ester (CFSE) and cultured alone or co-cultured with Daudi cells for a continuous 5 days. Cell proliferation was measured by CFSE dilution analysis by flow cytometry. The data showed that the MFI of CSFE in shACAT1-19CAR-T cells was similar to T, 19CAR-T, and NC-19CAR-T cells on day 0 (Figure 4a), however, a higher level of cell proliferation was observed in shACAT1-19CAR-T cells compared to T, 19CAR-T, and NC-19CAR-T cells on day 5 (Figure 4b). Similar results were also obtained in co-cultured cells (Figure 4c,d), which shows that silencing of ACAT1 enhanced the cell proliferation of CD19-CAR-transduced T cells in an antigen-independent or antigen-dependent manner.

### 3.5. Silencing of ACAT1 Gene Improves In Vivo Anti-tumor Activity of CD19-CAR-T Cells

To assess the in vivo functionality of CD19-CAR T cells with ACAT1 silencing, we used the NSG mouse tumor model, in which the animals were engrafted subcutaneously with 3 × 10^5^ Luciferase^+^ Raji cells and, after 7 days, mice were infused intravenously with three different groups of CD19-CAR-transduced T cells (5 × 10^6^ cells/mouse) (Figure 5a). All the CD19-CAR-transduced T cells demonstrated potent anti-tumor activity with a more than 60% survival rate, whereas, the CD19-CAR group with ACAT1-shRNA sets showed a more than 80% survival rate in xenograft B-cell leukemia mice model as shown in Figure 5b. Localization and expansion of the tumor were measured weekly using the Xenogen-IVIS imaging system. Representative mice are shown in Figure 5c, with the intensity of bioluminescence representing the tumor size. The mice from the CD19-CAR-T and NC-19CAR-T groups had less bioluminescence compared to the control group, in which the mice were treated with only PBS. Interestingly, shACAT1-19CAR-T cells worked more efficiently and reduced the tumor growth more than 19CAR-T and NC-19CAR-T cells, and the tumors in two mice were undetectable. Taken together, all the data showed that silencing the ACAT1 gene could increase the efficiency of CD19-CAR in eradicating B-cell lymphoma.

## 4. Discussion

Chimeric antigen receptor (CAR) expressed on T lymphocytes enable them to specifically target a wide range of human malignancies including Hodgkin and non-Hodgkin lymphomas [20,21,22,23,24]. CARs that target CD19 [25,26] in B-cell lineage have been cloned and authenticated in preclinical lymphoma/leukemia models and a few are currently in phase 1 clinical trials [20,22,24,27]. The CD28 co-stimulatory domain when inserted into CARs can enhance the activation of T cells [28,29,30,31]. However, the anti-tumor effects of CD19-based CAR T cells remain insufficient. An alternate domain, 4-1BB, has been proven to be superior to CD28 in certain in vivo studies that indicate CARs having CD137 have improved anti-leukemic efficacy and improved persistence in the primary human lymphoblastic leukemia xenograft model. CD137 provides anti-tumor efficacy that is antigen-independent and can contribute to the improved efficacy of CAR [13]. Therefore, the anti-tumor effects of CD19-based CAR systems remain limited for several reasons [32].

Recently metabolic regulation of T cells has been studied extensively because T cells undergo abnormal metabolic alterations due to the deleterious effects of tumor rendering them a wasted weapon [33]. Cholesterol metabolism has emerged recently as an important regulator of T-cell function with a broad-spectrum impact ranging from chemotaxis to cell-cycle progression and effector functions [34,35,36]. In response to cancer, when T cells change their state from naïve to activated, they readily reprogram their cholesterol metabolism with the help of SREBP and LXR transcription factors to promote cholesterol biosynthesis and decrease cholesterol transport out of the cell [34,35]. In addition to several other roles, free cholesterol clusters at T-cell receptors (TCRs) play a critical role in TCR signaling and T-cell activation [37,38,39]. ACAT1 is a key cholesterol esterification enzyme that converts free cholesterol into cholesterol esters, hence leaving less free cholesterol. Xu et al. recently demonstrated that inhibition of ACAT1 can potentiate CD8^+^ T-cell effector function. ACAT1 inhibitor Avasimibe, originally developed for the treatment of atherosclerosis, has been proven to be useful for treating lung cancer and melanoma. In the present study, while performing a cytotoxic assay, CD19-CAR-transduced T cells could significantly kill CD19-positive Raji cells compared with non-transduced T cells. In the current study, we have utilized shRNA technology to achieve ACAT1 downregulation in CD19-CAR-transduced T cells to enhance CAR-T cell therapy efficacy. Although both shRNA technology and CRISPR-Cas9 are powerful tools used for gene manipulation, we have utilized shRNA since some studies have mentioned that shRNAs are coupled with fewer off-target effects compared with CRISPR-Cas9. Additionally, we have applied shRNA technology previously against solid tumors and have obtained promising results [40]. Interestingly, CD19-CAR T cells with silencing of ACAT1 had a stronger killing ability. So here, we demonstrated that the killing capacity of CD19-CAR-transduced T cells can be enhanced by the silencing of ACAT1 gene expression.

The expression of CD69 is upregulated on a large number of leukocytes upon antigen stimulation, which is why it is widely regarded as an early activation marker of T lymphocytes as well as NK cells. CD69 is expressed on the surface of T lymphocytes soon after TCR/CD3 interaction as soon as within 30–60 min [41]. In addition to its value as a marker of T-cell activation, recently, it has been regarded as an important regulator of immune responses. It may determine the release of cytokines as well as the migration of activated T cells [42]. After T-cell activation, IFN-γ is one of the important cytokines known to possess immunoregulatory and anti-tumor properties [43]. Here, the downregulation of ACAT1 led to increased expression of both CD69 and IFN-γ. This means that ACAT1 interference through shRNAs led to enhanced activation of T cells and an upregulated expression of IFN-γ, which may be the reason for the increased killing capacity of CD19-CAR-transduced T cells. Similarly, the results are consistent with the data reported by Xu et al., who used an ACAT1 inhibitor to assess its impact on IFN-γ release [17].

Soon after activation of the T cells, degranulation takes place as a result of TCR activation. The major killing pathway of target cells by T cells occurs via a perforin/granzyme-dependent manner [44]. The core of lytic granules is composed of numerous proteins including perforins and granzymes [45]. The core is surrounded by lysosomal-associated membrane glycoproteins (LAMPs), including CD107a (LAMP-1). LAMP-1 is a marker of degranulation, which is important for perforin-mediated killing. Monitoring the level of granzymes alone is meaningless because it does not provide information about the degranulation capacity of cytotoxic T lymphocytes [46]. Our study demonstrated increased levels of both LAMP-1 and GzmB as a result of ACAT1 downregulation. We suppose that the increased degranulation of cytotoxic granules as well as upregulated expression of cytotoxic granzymes i.e., granzyme B, could be achieved in CD19-CAR T via ACAT1 interference.

Furthermore, we performed an in vivo experiment in the NSG mouse model. CD19-CAR T cells with ACAT1 interference led to better treatment results compared with conventional CD19-CAR T cells, the tumor was undetectable and the survival rate was highest in the shACAT1-19CAR T group.

While CD19 CAR-T cell therapy has shown remarkable efficacy in treating certain hematologic malignancies, it can also be associated with immunotoxicity, including on-target and off-target effects. We believe that while studying CD19-CAR-T cell products, considering the immunotoxicity factor is of vital importance and our study lacks these data, which is a limitation of our presented work and we intend to carry out detailed immunotoxicity studies within novel CAR-T cells presented herein.

## 5. Conclusions

In summary, the current study revealed that silencing of ACAT1 enhanced the activation and cell proliferation of anti-CD19-CAR T cells, and the capacity of anti-B-cell lymphoma was also improved both in vitro and in vivo. Our data demonstrated a novel CAR-T cell type with silencing of the ACAT1 gene had stronger anti-tumor efficacy compared to conventional CAR-T cells.

## Figures and Tables

**Figure 1 cells-13-00555-f001:**
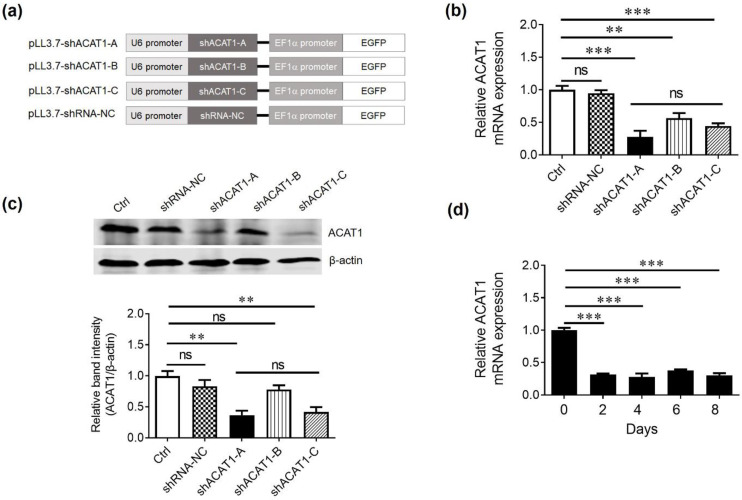
ACAT1 shRNA can effectively downregulate the expression of ACAT1 in Jurkat cell lines. (**a**) Schematic diagram showing the ACAT1-shRNA gene expression under U6 promoter in lentiviral vector pLL3.7. (**b**,**c**) Relative ACAT1 expression analysis in Jurkat cell line. Jurkat cells were transduced with lentiviruses expressing ACAT1-shRNAs and the relative gene and protein expression was confirmed by qPCR (**b**) and Western blot analysis (**c**). (**d**) The constant interfering ability of shACAT1-A was analyzed by qPCR with 2-day intervals until 8 days after transduction in Jurkat cells. GAPDH was used as an internal normalization control. Results are representative of three independent experiments. For all panels, the bars represent the mean ± SD. ns, not significant, ** *p* < 0.01, *** *p* < 0.001, ns, not significant.

**Figure 2 cells-13-00555-f002:**
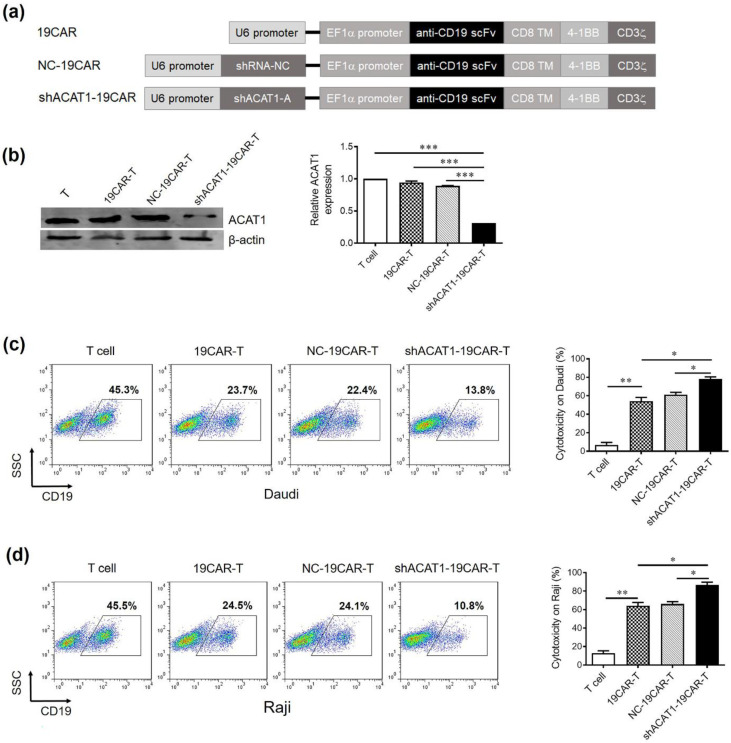
Silencing of the ACAT1 gene enhances the anti-tumor effect of CD19-CAR-transduced T cells. (**a**) Schematic diagram showing CD19-CAR (19CAR), CD19-CAR with ACAT1-shRNA (shACAT1-19CAR), and CD19-CAR with negative control shRNA (NC-19CAR). (**b**) The significant downregulation of the ACAT1 gene in CD19-CAR T cells was analyzed by Western blot (**left panel**). The relative expression level of ACAT1 was statistically analyzed (**right panel**). β-actin was used as an internal normalization control. (**c**,**d**) Cytotoxic activity of control, CD19-CAR, NC-CD19-CAR, or shACAT1-CD19-CAR-transduced T cells. All the T cells (effector cells) were mixed with Raji cells or Daudi cells at the E:T ratio of 1:1 for 16 h and CD19-positive cells were evaluated by flow cytometry. The representative pseudocolor plots are shown on the left side and the cytotoxicity rate was statistically analyzed, as presented on the right side. Results are representative of three independent experiments. * *p* < 0.05, ** *p* < 0.01, *** *p* < 0.001.

**Figure 3 cells-13-00555-f003:**
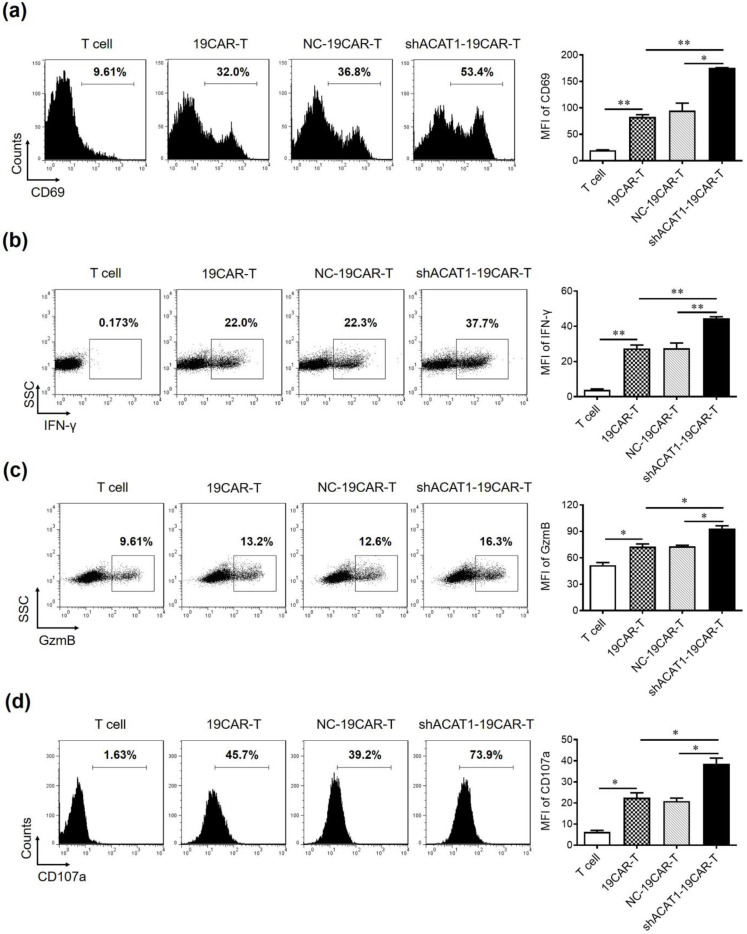
Downregulation of **the** ACAT1 gene improves the cell activation and degranulation of CD19-CAR-transduced T cells. (**a**–**c**) CD19-CAR-transduced T cells were co-cultured with Raji cells at the E:T ratio of 1:1 and the expression of (**a**) CD69, (**b**) IFN-γ, (**c**) GzmB, and (**d**) CD107a was evaluated by flow cytometry. Transduced T cells were stained with anti-human CD69-APC, anti-human IFN-γ-APC, anti-human GzmB-PE/Cy7, and anti-human CD107a-PE/Cy7 antibodies in separate experiments and analyzed by flow cytometry. The mean fluorescent intensity (MFI) of these proteins was statistically analyzed and is shown in the column chart. One-way ANOVA was used to measure the statistical significance indicated as * *p* < 0.05, ** *p* < 0.01.

**Figure 4 cells-13-00555-f004:**
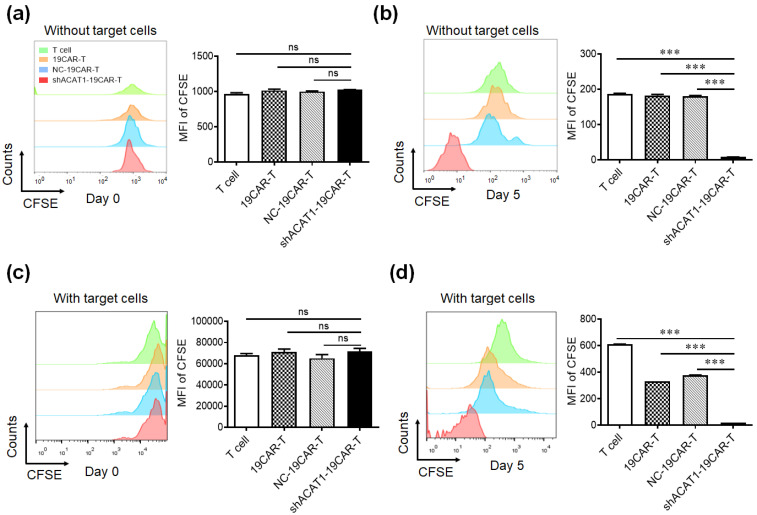
Downregulation of **the** ACAT1 gene improves the cell proliferation of CD19-CAR-transduced T cells. (**a**,**b**) CD19-CAR-transduced T cells or non-transduced T cells were stained with 2 μM/mL CFSE for 10 min and cultured for 5 days. CFSE dilution was analyzed by flow cytometry on day 0 (**a**) and day 5 (**b**), respectively. (**c**,**d**) CD19-CAR-transduced T cells or non-transduced T cells were stained with 2 μM/mL CFSE staining for 10 min and co-cultured with Daudi cells for 5 days. CFSE dilution was analyzed by flow cytometry on day 0 (**c**) and day 5 (**d**), respectively. The MFI of CFSE was statistically analyzed. Data are representative of three different experiments. *** *p* < 0.001, ns, not significant.

**Figure 5 cells-13-00555-f005:**
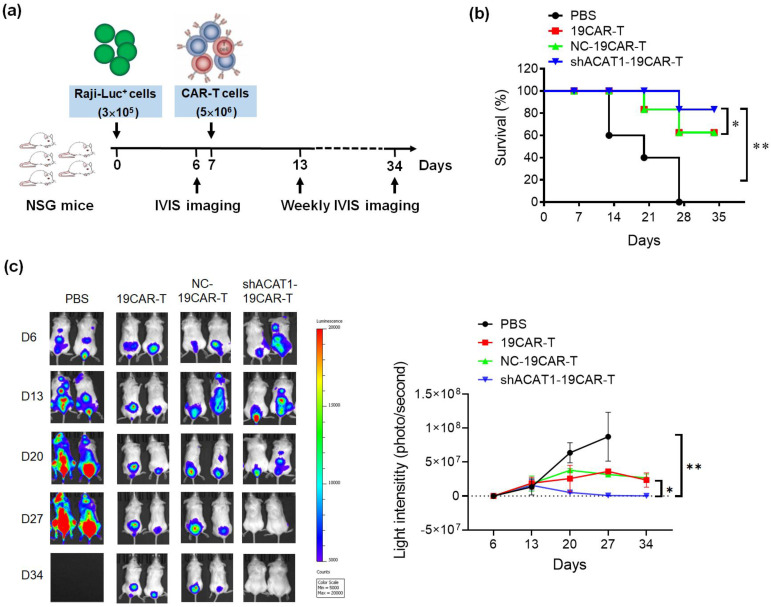
Silencing of ACAT1 enhances in vivo anti-B-cell lymphoma of CD19-CAR-transduced T cells. (**a**) Schematic diagram of in vivo experiment using NSG mice model. NSG mice (5 mice/group) were infused subcutaneously with FFLuc-labeled Raji cells and mice were i.v. injected with PBS, CD19-CAR T (19CAR-T), NC-CD19-CAR T (NC-19CAR-T), or shACAT1-CD19-CAR T (shACAT1-19CAR-T) cells (**b**) on day 7, Kaplan–Meier survival analysis of Raji-challenged mice (n = 3) after treatment with CD19-CAR-transduced T cells. (**c**) IVIS imaging (n = 2) was performed to monitor tumor burden at days 6, 13, 20, 27, and 34. The bioluminescence of tumor cells was measured and the light intensity (p/s) was analyzed. * *p* < 0.05, ** *p* < 0.01.

## Data Availability

The data that support the findings of this study are available from the corresponding author upon reasonable request.

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
