# Peer review of "Modulating Cholesterol Metabolism via ACAT1 Knockdown Enhances Anti-B-Cell Lymphoma Activities of CD19-Specific Chimeric Antigen Receptor T Cells by Improving the Cell Activation and Proliferation"

_cells, 2024, doi:10.3390/cells13060555_

Round 1

Reviewer 1 Report

Comments and Suggestions for Authors

This is an interesting pre-clinical study.

Line 32-33: it is not correct "Chemoimmunotherapy has remained the main stay of treatment 32 and as a first line treatment in 50-60% patients".

All patients with aggressive lymphoma who can tolerate chemotherapy must be treated with chemoimmunotherapy.

Author Response

Dear reviewer,

Thank you so much for your encouraging comments. 

Reviewer’s comment: Line 32-33: it is not correct "Chemoimmunotherapy has remained the main stay of treatment and as a first line treatment in 50-60% patients". All patients with aggressive lymphoma who can tolerate chemotherapy must be treated with chemoimmunotherapy.

Author’s response:We are extremely grateful to you for pointing out a valid error. We have indeed deleted the lines and added some additional information. The revised changes pertinent to the comment have been highlighted in yellow in the revised version.

Reviewer 2 Report

Comments and Suggestions for Authors

Manuscript explores the potential of targeting Acetyl-CoA acetyltransferase 1 (ACAT1), a key enzyme in cholesterol metabolism, to enhance the efficacy of CD19-specific CAR-T immunotherapy for B-cell lymphoma. By employing RNA interference to knock down ACAT1 in anti-CD19 CAR-T cells, the study demonstrates increased cytotoxicity, enhanced T cell activation markers, improved proliferation, and enhanced antitumor efficacy in a mouse model of B-cell lymphoma. These findings suggest that targeting ACAT1 could be a promising strategy in optimizing CAR constructs for the B-cell lymphoma therapy development.

Manuscript is well written, but may benefit from punctuation check and minor revisions:

Line 35: «Recently, two second generation chimeric antigen receptor (CAR) T cell therapy products tisagenlecleucel and axicabtageneciloleucel (axi-cel) targeting CD19 antigen have been approved by FDA for B cell lymphoma» – Actually, three products have been approved, including Lisocabtagene maraleuc (Breyanzi) for adult patients with LBCL;

Line 47: «In-vitro studies» - no need for dash https://dictionary.cambridge.org/dictionary/english/in-vitro;

Line 94: «After PBMC was isolated from 200 ml cord blood by using the density centrifugation» - clarify whether Ficoll, Lymphoprep-based, or similar methods were used.

Line 131, 138 and similar: « 1×106 Jurkat or primary T cells were seeded» - avoid starting sentences with numbers;

Materials and methodsPlease carefully review the information; suppliers, concentrations, and protocol details are missing.

Figure 5c- Please include photos of all five mice per group.

Discussion. The potential impact of shACAT1-mediated proinflammatory cytokine secretion on CAR-T cell immunotoxicity is one of the key issues and should be addressed in the discussion.

Comments on the Quality of English Language

Manuscript is well written, but may benefit from punctuation check

Author Response

Dear reviewer,

We are grateful for the positive encouragement you offered and are particularly thankful for the clarity with which you communicated your thoughts. Your dedication to the peer-review process is evident. We are committed to incorporating your suggestions diligently, and believe that your feedback will significantly contribute to the overall strength and credibility of our work.

Reviewer’s comment: Recently, two second generation chimeric antigen receptor (CAR) T cell therapy products tisagenlecleucel and axicabtageneciloleucel (axi-cel) targeting CD19 antigen have been approved by FDA for B cell lymphoma» – Actually, three products have been approved, including Lisocabtagene maraleuc (Breyanzi) for adult patients with LBCL;

Author’s response: You are absolutely right in pointing out the above-mentioned information. The new information has been incorporated in the revised version, and the information has been highlighted.

Reviewer’s comment: «In-vitro studies» - no need for dash https://dictionary.cambridge.org/dictionary/english/in-vitro;

Author’s response: Thank You for raising a valid typing error. The corrections have been made.

Reviewer’s comment: After PBMC was isolated from 200 ml cord blood by using the density centrifugation» - clarify whether Ficoll, Lymphoprep-based, or similar methods were used.

Author’s response: We used Ficoll-Paque PLUS density centrifugation method for PBMC isolation. Furthermore, we used CD3 magnetic beads to isolate the T cells. The specific protocol has been added in the materials and methods section under T cell sorting and activation headline and highlighted.

Reviewer’s comment: 1×106 Jurkat or primary T cells were seeded» - avoid starting sentences with numbers;

Author’s response: The problem has been solved.

Reviewer’s comment: Please carefully review the information; suppliers, concentrations, and protocol details are missing.

Author’s response: The information about the supplier information have been added. The protocols have been improved for the clarity, and have been expanded as suggested adding specific details. Please see highlighted sections in materials and methods section.

Reviewer’s comment: Figure 5c- Please include photos of all five mice per group.

Author’s response: Thanks for your kind suggestion. We have indeed used five mice per group for survival analysis, only two out of five mice were subjected to IVIS imaging for tumor growth analysis, because the IVIS imaging could cause additional stress (anesthesia and exposure to IVIS). That is why two mice can be seen in the Figure 5C.

Reviewer’s comment: Discussion. The potential impact of shACAT1-mediated proinflammatory cytokine secretion on CAR-T cell immunotoxicity is one of the key issues and should be addressed in the discussion. 

Author’s response: You are absolutely right about toxicity issue with CD19 targeting CAR-T cell products. We have added some information about the issue you have mentioned in the discussion part and these information has been highlighted.

Reviewer 3 Report

Comments and Suggestions for Authors

The work is well written, innovative and opens up new possibilities for acting on the enhancement of CAR T.

The experiments are well described and reproducible.

The simplicity of the idea, although demonstrated in a complex way, is probably the strong point of this paper together with the possibility of moving to a clinical trial in a short time.

I recommend the publication

Author Response

Dear reviewer,

I am thrilled to learn that you found the work well-written, innovative, and foreseeing new possibilities for advancing CAR T. Your acknowledgment of the clarity in describing the experiments and their reproducibility is especially encouraging. Your recognition of the simplicity of the idea, presented in a complex yet comprehensible manner, is truly uplifting. I am grateful that you see the potential for the research to swiftly transition to a clinical trial, underlining the practical impact of the work. Your recommendation for publication is a significant honor and affirmation of the effort invested in this project. I am sincerely thankful for your support and positive evaluation, which will undoubtedly inspire me in future research endeavors.

Reviewer 4 Report

Comments and Suggestions for Authors

Su et al. show that knock down of ACAT1 with shRNA has a beneficial influence on CAR T cell functionality both in vivo and in vitro.These results are in concordance with earlier reports. As a result the novelty lies in the use of shRNA instead of CRISPR/Cas9. Why did the authors decide to go with shRNA? If complete KO of ACAT1 is preferred, why not go for CRISPR/Cas9? Silencing of shRNAs is an often observed problem with shRNA.

Is it possible to show the entire western blot figures? Why is there background on the first ACAT1 blot but not on the second ACAT1 blot?

Figure 1: shRNA-C shows medium reduction in mRNA but the greatest reduction on protein level. The authors should elaborate more on the selection of shRNA-A and not C. In addition only the shRNA-A was tracked in time. Were the other shRNAs also followed? Is 8 days sufficient to report stable knockdown? Lastly, the legend of 1d doesn't pare with line 183-184. There is no mention of biological or technical replicates.

Figure 2: Where the other shRNAs tested on CAR T cells? Loss of CD19 on the target cells could be  caused by CAR-CD19 interactions. Did the authors assessed other markers that are independent of CAR-target cell interactions such as CD3? In addition, is six hours sufficient to measure cell death by FACS? There is no mention of biological or technical replicates.

Figure 3: Why did the authors measure CD69 so early? Figure 3b-c should be plotted using histograms or contourplots or pseudocolorplots. Figure 3a-d the y-axis on histograms is not SSC. Where the CAR T cells used in these experiments pure CD8 T cells or was it a mixture of CD4/CD8. This is important information, if it is a mixture of CD4/Cd8 T cells it could interfere with the Gzmb staining. These data should be shown in supplementals. In addition, the authors should make a comment in the text about which T cell subset was used.

Figure 4: How are the cells able to proliferate without the presence of target cells? The x-axis of the histograms are not the same. Please change this.

Line 81: On which basis were the shRNAs selected? In addition, the sequence of the shRNAs should be shown.

Line 145: both actin and GAPDH were used

Line 153: IVIS imaging was performed weekly, why mention "every two weeks" in M&M?

Line 184: 8 days for shRNA is quite short. Did the authors follow the Jurkat cells for a longer time period?

Comments on the Quality of English Language

Quality of English should be improved. Words seem to be missing or are misplaced.

Author Response

Dear reviewer,

Thank you so much for your time and a detailed comments. Here is a point-by-point response.

Reviewer’s comment: Su et al. show that knock down of ACAT1 with shRNA has a beneficial influence on CAR T cell functionality both in vivo and in vitro. These results are in concordance with earlier reports. As a result, the novelty lies in the use of shRNA instead of CRISPR/Cas9. Why did the authors decide to go with shRNA? If complete KO of ACAT1 is preferred, why not go for CRISPR/Cas9? Silencing of shRNAs is an often-observed problem with shRNA.

Author’s response: Thanks for your kind suggestions. Actually, both shRNA technology and CRISPR-Cas9 are powerful tools used for gene manipulation. We have utilized shRNA due to the fact that some studies have mentioned that shRNAs are coupled with fewer off-target effects as compared with CRISPR-Cas9. Additionally our laboratory is quite familiar with this technology and have applied shRNA technology previously against solid tumors and have got promising results (https://www.nature.com/articles/s41389-021-00353-8 ). Apart from excellent efficacy, establishing an shRNA-based system can be more cost-effective and efficient than setting up a CRISPR-Cas9 system. Because we can knockdown the gene expression with shRNA and express CAR in T cells at the same time if we use shRNA technology. However, we must knockout the interest gene in T cells with CRISPR-Cas9 system firstly, then we can transfect these T cells with lentivirus expressing CAR gene if we use CRISPR-Cas9 system, and we can not make sure gene knockout and CAR expression in the same cells.

Reviewer’s comment: Is it possible to show the entire western blot figures? Why is there background on the first ACAT1 blot but not on the second ACAT1 blot?

Author’s response: Thanks for your question. We have provided the entire western blot figures. Actually, we performed the two experiment in different time and the antibodies came from different company, the specificity of the first antibody was not good, so the background was different. We judged the band according to the protein marker.

Reviewer’s comments: Figure 1: shRNA-C shows medium reduction in mRNA but the greatest reduction on protein level. The authors should elaborate more on the selection of shRNA-A and not C. In addition, only the shRNA-A was tracked in time. Were the other shRNAs also followed? Is 8 days sufficient to report stable knockdown? Lastly, the legend of 1d doesn't pare with line 183-184. There is no mention of biological or technical replicates.

Author’s response: We appreciate for your kind suggestions. We have now added the relative band intensity graph of western blot analysis which you pointed out and we should have added that in the initial version to rule out the possibility of any confusion. The figure depicts that at protein level there was non-significant difference between shRNA-A and shRNA-C. The similar trend can be seen in qPCR data. We selected shRNA-A due to its higher GC content. The literature suggests that higher GC content provides stable secondary structure and effective interaction with target mRNA.

Other shRNAs were not tracked for extended period of time for stable downregulation.

Because most of our in vitro experiments were less than 8 days of coculture, so we have analyzed downregulation till 8 days, which is sufficient for current study. 

You are right about our statement in the Figure and it has been revised. Please see the highlighted text.   

Reviewer’s comments: Figure 2: Where the other shRNAs tested on CAR T cells? Loss of CD19 on the target cells could be caused by CAR-CD19 interactions. Did the authors assessed other markers that are independent of CAR-target cell interactions such as CD3? In addition, is six hours sufficient to measure cell death by FACS? There is no mention of biological or technical replicates.

Author’s response: No, we didn’t test other shRNAs in CAR-T cells but tested shRNA-A. We only assessed CD19 on target cells as a measure of cytotoxicity because the target cells such as Daudi or Raji could express CD19. We believe that the antigen shedding does not take place in vitro due to short time period of coculture. It was actually 16 hours and not 6 for the killing time, the typing error has been corrected and highlighted.

Reviewer’s comments: Figure 3: Why did the authors measure CD69 so early? Figure 3b-c should be plotted using histograms or contourplots or pseudocolorplots. Figure 3a-d the y-axis on histograms is not SSC. Where the CAR T cells used in these experiments pure CD8 T cells or was it a mixture of CD4/CD8. This is important information, if it is a mixture of CD4/Cd8 T cells it could interfere with the Gzmb staining. These data should be shown in supplementals. In addition, the authors should make a comment in the text about which T cell subset was used.

Author’s response: Thanks for your kind suggestions. CD69 is an early activation marker. According to the literature, CD69 is expressed even 30 minutes after antigen encounter.

Fig 3b-c, the populations can be differentiated in a clearer manner. That is why, data has been provided in dot plots.  

Fig 3a and 3d, Y axis is not SSC but counts, the mistake was corrected where applicable.

For CAR-T cells in current research, it was a mixture of CD4 and CD8+T cells.

Comment has been added in the results section for further clarity.

Reviewer’s comments: Figure 4: How are the cells able to proliferate without the presence of target cells? The x-axis of the histograms are not the same. Please change this.

Author’s response: Thanks for your questions. Actually, we stimulated the T cells with CD3/CD28 antibodies, which was the non-specific stimulation (Fig.4b), the target cells were the specific stimulation (Fig.4d). According to the available literature, ACAT-1 gene deficiency can enhance proliferation of T cells by increasing cell activation potential (https://www.nature.com/articles/nature17412 and https://www.ncbi.nlm.nih.gov/pmc/articles/PMC7063140/ ). So in our study, the increased proliferative potential of ACAT1 downregulated CAR-T cells might correspond to enhanced proliferative potential as a result of improved cell activation.

Reviewer’s comments: Line 81: On which basis were the shRNAs selected? In addition, the sequence of the shRNAs should be shown.

Author’s respons: shRNAs were selected on these criteria. (1) the sequence should lie within open reading frame region to get direct translational interference (2) 5’ end shall start with G nucleotide (3) GC content should be between 30-50%. (4) there should not be 3 or more Thyamines. This information along with shRNAs sequences have been added and highlighted. Please see materials and methods section.

Reviewer’s comments: Line 145: both actin and GAPDH were used

Author’s response; Thanks for your reminding. We have added the information of GAPDH and it was highlighted.

Reviewer’s comments: Line 153: IVIS imaging was performed weekly, why mention "every two weeks" in M&M?

Author’s response: That was a typing error and has been rectified.

Reviewer’s comments: Line 184: 8 days for shRNA is quite short. Did the authors follow the Jurkat cells for a longer time period?

Author’s response: Thanks for your question. We regret we have not followed the Jurkat cells for a longer time period. Actually, we detected the biological function less then 8 days and we could detect the enhanced function in ACAT1-knowdown CAR-T cells. Therefor, 8 days are enough for downregulation efficiency in current study. 

Reviewer’s comments: Quality of English should be improved. Words seem to be missing or are misplaced.

Author’s response: Thanks for your kind suggestions. We have checked the manuscript and made a revision.

Round 2

Reviewer 4 Report

Comments and Suggestions for Authors

I thank the authors for addressing my questions.

I think we can have a long debate on the use of CRISPR/Cas9 vs shRNA. I acknowledge the answer of the author. Perhaps this answer could be mentioned in the introduction or the discussion? I leave this up to the editor/authors to decide.

Western blot: This is crucial information, perhaps I missed it in the M&M but this should be mentioned in M&M.

Fig 1. Thank you for the incorporation of the extra sub figure. However I still miss the number of biological or technical replicates.

Fig 2. Antigen shedding can occur even after 1 hour (see PMID: 30918399). Another marker would have been a  better choice but I understand that a redo of the experiments could take a longer time. Thank you for the addition of the number of experiments.

Fig 3b-c. One could argue about the use of dot plots. My preference goes to pseudocolor or contourplots. I leave this to the editor to decide which to show. Were the ratios of CD4/CD8 cells determined over the different conditions? If so where they the same?

Fig 4. Thank you for this clarification. Perhaps mention this also in the manuscript? The X-axis is not yet altered? Could this still be done before publication of the manuscript?

Comments on the Quality of English Language

Some minor typos are still present.

Author Response

We are again thankful to the worthy reviewer for giving insightful comments and his/her keen interest in our current study. Here is a point-by-point response for the second round of review.

Reviewer’s comment:I think we can have a long debate on the use of CRISPR/Cas9 vs shRNA. I acknowledge the answer of the author. Perhaps this answer could be mentioned in the introduction or the discussion? I leave this up to the editor/authors to decide.

Author’s response:We are grateful to the worthy reviewer for acknowledging our reasoning. We have indeed taken account of your concern. For further clarity, the author’s initial response in this regard has been added in the discussion section.

Reviewer’s comment: Western blot: This is crucial information, perhaps I missed it in the M&M but this should be mentioned in M&M.

Author’s response:Thank you for this valuable suggestion. We have mentioned the information about antibodies in the Materials and Methods section under the Western Blot headline.

Reviewer’s comment:Fig 1. Thank you for the incorporation of the extra sub figure. However I still miss the number of biological or technical replicates.

Author’s response The information has been incorporated into the legend of Fig 1.

Reviewer’s comment: Fig 2. Antigen shedding can occur even after 1 hour (see PMID: 30918399). Another marker would have been a better choice but I understand that a redo of the experiments could take a longer time. Thank you for the addition of the number of experiments.

Author’s response:We are grateful for your understanding.

Reviewer’s comment:Fig 3b-c. One could argue about the use of dot plots. My preference goes to pseudocolor or contourplots. I leave this to the editor to decide which to show. Were the ratios of CD4/CD8 cells determined over the different conditions? If so where they the same?

Author’s response:We understand the importance of the reviewer’s remark. We have detected the impact of ACAT-1 downregulation on the CD8/CD4 T cell ratios in ACAT-1 downregulated NKG2D-CAR-T cells and found no difference in the absence of target cells (data not shown here). However, in co-culture conditions, the percentage of CD8+ CAR-T cells was significantly increased in ACAT-1 downregulated CAR-T cells compared to the wild-type CAR-T (data not shown here).  Although we didn’t carry out this analysis for CD19 targeting CAR-T cells in the current study, we believe the mentioned results can be extrapolated.

Reviewer’s comment:Fig 4. Thank you for this clarification. Perhaps mention this also in the manuscript? The X-axis is not yet altered? Could this still be done before publication of the manuscript?

Author’s response:Thanks for your kind suggesitons. We have supplemented some information in section 3.4 and the information has been highlighted. X-axis has been altered according to your suggestion.